# Primary Amine Functionalized Carbon Dots for Dead and Alive Bacterial Imaging

**DOI:** 10.3390/nano13030437

**Published:** 2023-01-21

**Authors:** Yuting Liu, Di Zhong, Lei Yu, Yanfeng Shi, Yuanhong Xu

**Affiliations:** 1Institute of Biomedical Engineering, College of Life Science, Basic Medical College, Qingdao University, Qingdao 266071, China; 2Department of Genetics and Cell Biology, Basic Medical College, Qingdao University, No. 308 Ningxia Road, Qingdao 266000, China

**Keywords:** carbon dots, bacteria, imaging

## Abstract

Small molecular dyes are commonly used for bacterial imaging, but they still meet a bottleneck of biological toxicity and fluorescence photobleaching. Carbon dots have shown high potential for bio-imaging due to their low cost and negligible toxicity and anti-photobleaching. However, there is still large space to enhance the quantum yield of the carbon quantum dots and to clarify their mechanisms of bacterial imaging. Using carbon dots for dyeing alive bacteria is difficult because of the thick density and complicated structure of bacterial cell walls. In this work, both dead or alive bacterial cell imaging can be achieved using the primary amine functionalized carbon dots based on their small size, excellent quantum yield and primary amine functional groups. Four types of carbon quantum dots were prepared and estimated for the bacterial imaging. It was found that the spermine as one of precursors can obviously enhance the quantum yield of carbon dots, which showed a high quantum yield of 66.46% and high fluorescence bleaching-resistance (70% can be maintained upon 3-h-irradiation). Furthermore, a mild modifying method was employed to bound ethylenediamine on the surface of the spermine–carbon dots, which is favorable for staining not only the dead bacterial cells but also the alive ones. Investigations of physical structure and chemical groups indicated the existence of primary amine groups on the surface of spermine–carbon quantum dots (which own a much higher quantum yield) which can stain alive bacterial cells visibly. The imaging mechanism was studied in detail, which provides a preliminary reference for exploring efficient and environment-friendly carbon dots for bacterial imaging.

## 1. Introduction

Bacteria, such as *Escherichia coli* (*E. coli*) and *Staphylococcus aureus* (*S. aureus*) [1,2,3,4], can threaten human health through occurring inflammation, histopathological destruction, etc. Accordingly, early recognition and diagnosis are important to inhibit bacterial infection in time. Among all monitoring methods, fluorescent imaging could give a fast and visual result for bacterial cells observation [5,6,7,8,9]. Small molecular dyes such as SYTO series or propidium iodide (PI) for bacterial imaging have been developed for commercial use [7,8,9,10,11]. However, aromatic structured small molecular dyes are usually toxic for both the human body and the environment and their fluorescent structure is easily destroyed or changed under fluorescent excitation light in a short period of time [10,12]. Therefore, environmentally friendly dyes that do not cause photobleaching still urgently need to be exploited. Because of the volume and structure difference between mammal cells and bacterial cells, staining bacterial cells is much harder than staining mammal cells, since bacteria are much smaller and bacterial cell walls are more compact and less permeable which requires a harsh standard for bacterial stain [13,14,15,16].

Carbon dots have been recently widely developed as promising fluorescent probes for biomedical imaging due to their advantages of low-cost, environmentally friendliness and photobleaching-resistance as well as their good biocompatibility and simple synthesis [17,18,19]. Until now, different types of carbon dots have been developed for potential bacterial imaging [13,20,21,22,23,24,25,26]. Precursors have been made of citric acid, small molecule organic amine, amino acids or bacteria themselves with one or two synthetic steps [22,23,25,26,27,28]. Some carbon dots can discriminate alive and dead bacterial cells [22,23], while some can alternatively stain gram-positive or gram-negative bacteria [29]. Generally, imaging mechanisms of some kinds of carbon dots for dead bacterial cells were explained as the small size of carbon dots. Since the negatively charged carbon dots have a electrostatic repulsion toward negatively charged cell walls of alive bacterial cells, the carbon dots cannot interact with alive ones [22,24,25]. Amidogen or amino-contained substances have been reported to be able to attach to bacterial surfaces whether located at carbon dots or antibiotics [30,31,32,33,34]. Accordingly, certain types of carbon dots were suggested to be able to mark alive bacteria due to the small size (which could go through the damage membrane [35]) or the protonated amino group on carbon quantum dots [13,21], but most work did not give a systematic study on the mechanism of carbon dots-based bacterial imaging. Moreover, despite the enhancement in the anti-bleaching property, the large space to enhance the quantum yield hindered the carbon dots reaching efficient bacterial imaging [36,37,38,39]. There are also other types of nanoparticles used as fluorescent probes for biomedical imaging such as metal-contained quantum dots, silicon dots or conjugated polymers. Metal-contained ones are usually applied for bacterial cell killing instead of for imaging and they are toxic to the environment [20]. Silicon dots need a higher synthesizing temperature than carbon dots [40,41]. Conjugated polymers do have strong light adsorption due to their large, conjugated planes, but the cost of monomer synthesis is high and the synthetic procedure is complicated [42,43].

Herein, we aim to clarify the mechanism of the surface chemical groups of amidogens on carbon dots for their staining behaviors in bacterial imaging. Citric acid and ethylenediamine were employed as precursors for preparing the carbon dots. Spermine, as an additional precursor, was discovered to be efficient in enhancing high quantum yield. Ethylenediamine were further modified on the surface of spermine-contained carbon dots using EDC/NHS (N-(3-dimethylaminopropyl)-N’-ethylcarbodiimide hydrochloride and N-hydroxysuccinimide) chemistry which is a mild modifying method under room temperature. The preparation of spermine-contained carbon dots and the fabrication of primary amine functionalized carbon dots are illustrated in Figure 1. Quantum yield and functional group sizes were studied as the influencing factors for bacterial imaging. Investigations of the physical structures and chemical groups indicated the existence of primary amine groups on carbon quantum dots that are favorable for staining not only the dead bacterial cells but also the alive ones.

## 2. Materials and Methods

### 2.1. Chemicals and Reagent

Citric acid, ethylenediamine·H_2_O, spermine tetrahydrochloride, sodium chloride, N-(3-dimethylaminopropyl)-N’-ethylcarbodiimide hydrochloride (EDC), N-hydroxysuccinimide (NHS), 2-(N-Morpholino)ethanesulfonic acid monohydrate (MES), glycine, 2-[Tris(hydroxymethyl)methylamino]-1-ethanesulfonic acid (TES), sodium bicarbonate, sodium carbonate, sodium hydroxide, hydrogen chloride, disodium hydrogen phosphate, sodium dihydrogen phosphate, acetic acid, sodium acetate, sulfuric acid, quinine sulfate dihydrate, 3-(4,5-dimethyl-2-thiazolyl)-2,5-diphenyltetrazolium bromide (MTT) and dimethyl sulfoxide (DMSO) were purchased from Macklin Co., Ltd. In addition, 3% benzalkonium bromide was purchased from Shandong Lierkang medical technology Co., Ltd. Yeast extract and tryptone were purchased from OXOID Ltd. (Basingstoke Hampshire, United Kingdom) *Escherichia coli* (*E. coli*, ATCC 25922) and *Staphylococcus aureus* (*S. aureus*, ATCC 25923) were purchased from Beyotime Ltd. L929 cell lines were obtained from the Institute of Cell Biology, Chinese Academy of Sciences.

### 2.2. Apparatus

Transmission electron microscopy (TEM) and high-resolution TEM (HRTEM) were obtained using a JEOL JEM-2100 microscope (JEOL, Japan) at an acceleration voltage of 80 kV or 200 kV, respectively. X-ray diffraction (XRD) characterizations were obtained using a Bruker D8 Advance diffractometer with a copper Ka (k = 0.154056 nm) radiation source. X-ray photoelectronspectroscopy (XPS) measurements were obtained using a PHI5000 Versaprobe-II spectrometer (ULVAC-PHI, Japan) with a monochromatic Al Ka (1486.6 eV) source. The UV absorption spectra were obtained with a UV-6300 spectrophotometer (Mapada instruments Ltd., Shanghai, China). The photoluminescence spectra were obtained using an FS5 spectrophotometer (Edinburgh instruments Ltd., Livingston, UK). The fluorescence images were obtained with an upright fluorescence microscope (i203 type, Chongqing UOP, Chongqing, China) and an Eclipse Ni-U (Nikon Ltd., Tokyo, Japan) fluorescence microscope. Fourier-transform Infrared Spectrometer (FTIR, Thermo Fisher Scientific Ltd., Waltham, MA, USA) was employed to carried out IR spectra. Atomic force microscope (AFM, Dimension Icon, Bruker Co. Ltd., USA) was used to obtain AFM images. The particle size distribution and zeta potential were studied by dynamic light scattering (DLS) using Malvern Zetasizer (Malvern Instruments Ltd., Malverm, UK).

### 2.3. Preparation and Modification of the Carbon Dots

For the carbon dots, 0.96 g citric acid, 1250 μL ethylenediamine·H_2_O and 25 mL ultra-pure water were added into a 50 mL Teflon reactor. The reactor was kept at 180 °C for 6 h with a step of 6 °C/min. After cooling down, the mixture was filtered with 0.22 μm microporous membrane twice. The filtrate was dialyzed (1000 or 500 Da) for 24 h under room temperature. Then, the purified colloidal solution underwent a freeze-drying process to obtain the carbon dots (CD). The CD which synthesized using spermine (spCD) as one of percursors was fabricated by similar process mentioned above expect for adding spermine tetrahydrochloride in the hydrothermal step. Both CD and spCD were stored at room temperature and kept away from light and moisture.

An EDC/NHS chemistry was selected to complete the modification, which is a mild method for surface modification under room temperature [44]. Usually, EDC/NHS chemistry is employed for protein combination through carboxyl–amidogen bonding [44]. In this work. Carboxyl on the carbon dots was bonded to the primary amine group on the ethylenediamine (ethylenediamine have two primary amine group at both end of the compound). Briefly, 5~10 mg/mL CD or spCD, 8 mg EDC, 8 mg NHS were added into 6 mL 0.1 mol/L MES buffer (pH = 6.0) under 37 °C for 1 h. The amount of EDC and NHS were expected to be superfluous. Then, 1 mL ethylenediamine was added. Subsequently, pH value was adjusted to around 10.6 [44]. After incubation for 2 h under 37 °C, the solution was dialyzed to reach the modified CD (we named them NH_2_–CD) or the modified spCD (we named them NH_2_–spCD) suspension. Subsequently, the modified ones were obtained by freeze-drying the suspensions.

### 2.4. Quantum Yield (QY) Measurement

The QY of carbon dots were calculated and measured by the relative quantum yield. Typically, quinine sulfate dihydrate was diluted under 0.1 mol/L sulfuric acid solution (QY = 0.54 under 345 or 350 nm excitation wavelength). The carbon dots suspensions were diluted in different concentrations whose UV–Vis absorbance was not allowed to be higher than 0.1. The excitation wavelength of CD (including NH_2_–CD) and spCD (including NH_2_–spCD) were 345 nm and 348 nm, respectively. Equation of QY were defined as follows [45]:$${Ф}_{x}={Ф}_{ST}\left(\frac{Grad_{x}}{Grad_{ST}}\right)\left(\frac{\eta_{x}^{2}}{\eta_{ST}^{2}}\right)$$
where the *x* and *ST* refer to carbon dots and quinine sulfate dihydrate separately and Ф represent QY. Grad refers to the slope of the integrated fluorescent intensity dividing absorbance and *η* refers to solvent refractive index.

### 2.5. Bacterial Culture and Preparation for Imaging

*E. coli* and *S. aureus* were cultured with liquid medium, containing 0.5 g tryptone, 0.5 g sodium chloride and 0.25 g yeast extract per 50 mL sterile water. Then the bacteria was cultured under 150 rpm overnight to reach a O.D. value of around 0.5. For bacterial culture with carbon dots, alive bacteria were just centrifuged and rinsed with the medium and then carbon dots suspension were added and cultured for 30 min (*S. aureus* for 2 h). The preparing methods of dead cells were in two ways. One was killing bacteria with 3% benzalkonium bromide for 3 min, the other was killing with 60 °C water bath for 1 h [22,24]. Then the dead bacteria were co-cultured with the corresponding carbon dots for 30 min before the imaging characterization.

### 2.6. Stability Study of the Carbon Dots

The stability experiments were carried out by measuring the fluorescent intensity change under constant ultraviolet irradiation, in different concentrations of saline solution (9 different concentration gradients of sodium chloride ranging from 0 to 2.4 mol/L). Instant stability experiment was performed by collecting the fluorescence signal 13 times without interval time and the fluorescence signals of each kind of carbon dots sample were tested every 15 min. While long-term stability experiment was evaluated once per week. Excitation wavelength of 365 nm were employed throughout the whole experiment.

### 2.7. Cytotoxicity Test

MTT assay was selected to perform cytotoxicity of carbon dots. L929 cells were employed as model cells. Dulbecco’s modified Eagle’s medium (DMEM) was used as a cultural condition at 37 °C with 5% CO_2_ in an incubator. In brief, L929 cells were cultured and added 100 μL (contain around 8000 cells) into a 96-well plate for each hole. After being incubated for 24 h, 20 μL MTT solution were added for each hole, subsequently incubated for extra 4 h. The medium was removed and 150 μL DMSO was added for a 10 min shaking. The untreated ones were regarded as the control group. Finally, the absorbance of Formazan was read at 492 nm. The relative cell viabilities (%) after treatment with carbon dots at different concentrations were calculated [46]:$$\mathrm{Relative}\;\mathrm{viability}\left(\%\right)=\frac{{\mathrm{OD}}_{\mathrm{sample}}}{{\mathrm{OD}}_{\mathrm{control}}}\times 100\%$$

## 3. Results and Discussion

### 3.1. Fluorescent Characterization of Carbon Dots

Biogenic amine is a kind of low-molecular weight organic amine, which can participate in regulating the content of nucleic acid or protein to influence cellular metabolism and membrane stabilization for many microbes [36,37,47,48]. In addition, spermine presents a cationic charge in physiological environments and contains primary and secondary amine which could be a good candidate for the synergistic combating of pathogenic bacteria and potential bacterial dyes [38,39,47,48]. Accordingly, spermine as a kind of biogenic amine was exploited as a precursor for the carbon dots preparation and modification herein.

Fluorescent properties including excitation–emission (ex–em) spectra, emission maps, fluorescence lifetime and quantum yields were firstly studied, respectively. As shown in Appendix A, the maximum excitation and emission of CD were 346 nm and 445 nm, respectively, which were accorded with NH_2_–CD, as shown in Appendix A. Similarly, as Figure 1a depicts, the maximum excitation and emission values of spCD were 348 nm and 445 nm, which were consistent with the ones of NH_2_–spCD (Appendix A). It can be observed that the modifying process did not affect the peak positions of CD and spCD. All types of carbon dots have similar excitation/emission trends. When the excitation wavelength is lower than 400 nm, there are no exciting-dependent phenomenon except for the intensity changes, which also indicated that these carbon dots may have a relative uniform size distribution. In addition, all types of carbon dots have a wide and strong fluorescent emission ranging from 400 nm to 600 nm, which indicated its application possibility for biological imaging and the 1931 CIE chromaticity images were given out as shown in Figure 1c and Appendix A. All carbon dots gave out a blue to green fluorescence, where the blue light is attributed to the carbon core and the green fluorescence should result from the nitrogen or oxygen contained auxochromic groups [49]. The calculation of QYs referring to CD, spCD, NH_2_–CD and NH_2_–spCD were 48.79%, 66.46%, 40.42% and 53.47%, respectively. Obviously, a dramatic QY enhancement (nearly 20%) was obtained after nitrogen doping using spermine, while it decreased slightly after modifying with ethylenediamine. In addition, the lifetime depicted at Figure 1d did not have significant difference among the four different carbon dots. Enhanced QY should occur due to the nitrogen doping, which has been confirmed in previous research [50,51]. Usually, the fluorescence partially depends on the carbon core that has a rigid aromatic structure, which is made up of the hybridization of sp^2^ and sp^3^ with a typical ratio. Moreover, the synergy between the carbon core and functional chemical groups turns out to improve the QY [52]. In addition, the stability of carbon dots [52] were investigated for sustained irradiation, a high concentration of ion strength and constant detection. Figure 1b shows fluorescent intensity changes for 3 h ultraviolet irradiation. All the relative intensities of the carbon dots were declined after irradiation to different extents. The CD left only 50% after irradiation while the spCD remained at 70%, which indicates that doping undertook a certain degree of resistance to photobleaching. As depicted in Appendix A, even 2.4 mol/L NaCl could not lead to a declining trend in the relative intensity of the four carbon dots. Figure 1e shows the intensity changes of the carbon dots for a constant detection (15 times) contrast to those of SYTO9 and PI, which again confirmed the strong and stable fluorescence of the four types of carbon dots. As shown in Appendix A, the fluorescent intensity of CD and spCD only declined to 80~90% after 8 weeks, verifying the reasonable stability of the four carbon dots. For the modified ones (NH_2_–CD and NH_2_–spCD), the fluorescent intensity has a slight increase starting from the sixth week, which indicates that the ethylenediamine may lose a part in the two types of carbon dots. By contrast, SYTO9 and PI showed worse stability under the same circumstance, where the peak of fluorescent intensity of SYTO9 could not be detected in the fifth week and the one with PI had a dramatically red shift after around one month and its intensity decreased with the ongoing time.

### 3.2. Physical Structural Analysis of Carbon Dots

Figure 2a shows the XRD characterization of carbon dots, which all showed a poor crystallinity of materials. Except for CD, spCD and the modified ones all have an obvious peak at the lattice plane of (004) located at 2θ = 32.291. As Figure 2c,d shows, HRTEM characterization exhibited the lattice spacing of around 0.294 nm and 0.298 nm, which is consistent with the JCPDS standard card (No. 46-0870, nitrogen oxide graphite). It confirmed the doping of nitrogen into the core structural between carbon chemical bonds. An average diameter of 2.01 nm and 1.38 nm for CD and spCD can be obtained, respectively. NH_2_–CD and NH_2_–spCD showed the average diameters around 2 nm (Appendix A). Obviously, the size of the modified carbon dots was not increased extremely after the EDC/NHS reaction. AFM images of the four carbon dots are also given out in Figure 2e and Appendix A, which indicates the uniformity of the size and the thickness of 1.01~2.99 nm. The TEM and AFM characterizations confirms the successful formation of the carbon dots, respectively. Diameters and relevant QYs from other works are exhibited in Figure 2b, confirming the smaller sizes and higher quantum yield compared with the previous works [13,15,16,22,26,27,28,40,41,53,54,55,56,57,58,59,60,61,62,63].

### 3.3. Chemical Structural Analysis of Carbon Dots

UV–Vis spectra, FTIR and XPS characterizations were employed to analyze the chemical structure of carbon dots. As shown in Figure 3a, four carbon dots have a similar UV–Vis peak distribution. Therein, the peak located at 230~260 nm represents a π→π* transition which means the existence of chromophore such as sp^2^ hybridization, while the peak located at 300~400 nm represents an n→π* transition which means the existence of auxochrome including C–N, C–O or C=O. Neither the addition of spermine nor the EDC/NHS modification of ethylenediamine had largely changed the UV–Vis spectra. FTIR and X PS were further carried out to identify the surface chemical groups and elemental compositions. As depicted in Figure 3b, all types of carbon dots own a carboxyl characteristic wide peak at 3500~2500 cm^−1^ and all have alkane stretching vibration peaks (e.g., 2928 and 2863 cm^−1^ for NH_2_–spCD). The distribution of the peaks was identical between CD and spCD as well as between NH_2_–CD and NH_2_–spCD. Comparing with CD and spCD, the double peaks appeared at 3370 and 3273 cm^−1^ for NH_2_–CD and NH_2_–spCD, which may represent the amino stretching vibration. The peak at 1648 cm^−1^ may be the amide I band or C=O stretching vibration, while a secondary amide II band may located at 1548~1529 cm^−1^ [37]. For NH_2_–CD and NH_2_–spCD, 1382 and 1308 cm^−1^ may represent a C–N stretching vibration for aromatic primary ammonia and aromatic secondary ammonia. For CD and spCD, 1384, 1354, 1321 and 1219 cm^−1^ also represent aromatic ammonia spCD [34]. Peaks at 1214, 1182 or 1215 cm^−1^ may represent a C–O stretching vibration for the four types of carbon dots [64]. Peaks at 1648, 1548~1529, 1382 and 1308 cm^−1^ also suggest the formation of an aromatic ammonia structure, which is consistent with the UV–Vis results indicating the existence of sp^2^ hybridization. Usually, when aromatic structures are formed, most peaks may have a red shift. For example, a C=O stretching vibration should be located at around 1735 cm^−1^. However, in this case, a C=O stretching vibration was located at around 1648 cm^−1^ which matched the characteristic peak of amide. Differing from the red shift condition, peaks of an N–H stretching vibration would have a blue shift if the aromatic structure appeared. As a result, the double peaks at 3370 and 3273 cm^−1^ are most likely the primary amine characteristic peak [65]. Based on these characterizations, the peak at 932 cm^−1^ may be the N–H out-of-plane bending vibration that CD and spCD did not have, which further suggests the existence of amino on the NH_2_–spCD and NH_2_–CD [34,64].

Figure 4 shows the full-scan and high-resolution of XPS spectra for CD and spCD, respectively. In Figure 4a, the three peaks located at 284.8 eV (62.34%), 399.84 eV (9.39%) and 531.93 eV (28.27%) correspond to C 1s, N 1s and O 1s, respectively, for CD. Similarly, the three peaks located at 284.8 eV (65.18%), 399.93 eV (15.99%) and 531.99 eV (18.82%) correspond to C 1s, N 1s and O 1s, respectively, for spCD (Figure 4b). Upon the spermine being applied, the content of nitrogen increased while the content of oxygen decreased, which suggests the spermine structure had combined with the citric acid and ethylenediamine while the high-resolution XPS of both CD and spCD presented similar chemical circumstances for each element. For instance, as Figure 4f depicts, the C 1s spectrum of spCD (Figure 4f) shows C–C (284.46 eV), C–N (285.2 eV), C–O (285.95 eV) and C=O (287.57 eV) [45,66]. The N 1s spectrum of spCD (Figure 4j) shows C=N (399.36 eV), C–N (400.07 eV) and N–H (401.04 eV) [65,66]. The O 1s spectrum of spCD (Figure 4n) shows C–O (530.50 eV), C=O (531.13 eV) and O–H (532.30 eV) [37]. The QY enhancement upon the spermine addition should be due to the enhanced nitrogen content and rich pyridine nitrogen in the as obtained spCD, since pyridine nitrogen can increase defect states to create defect emission on the base of intrinsic emission when excited [67]. Meanwhile, in the full-scan and high resolution XPS spectra of NH_2_–CD (Figure 4c,g,k,o) located at 284.8 eV, 399.79 eV and 531.82 eV, and XPS sepctra of NH_2_–spCD (Figure 4d,h,l,p) peaks located at 284.8 eV, 399.41 eV and 5531.67 eV correspond to C 1s, N 1s and O 1s, respectively [24,45,65,66,68]. The results indicated that NH_2_–CD and NH_2_–spCD have similar chemical structures according to the high-resolution XPS spectra. However, one point should be noticed that peaks of 399.70 eV (C–N or −NH_2_) and 401.60 eV (N–H) may mean the existence of amino in N 1s spectra for NH_2_–CD, and 399.73 eV (C–N or −NH_2_) and 402.18 eV (N–H) for NH_2_–spCD [24,37]. On the contrary, CD and spCD did not have the identical peaks. Herein, FTIR spectra analysis has already suggested that the modified (ethanediamine using EDC/NHS chemistry) ones have amino groups while the non-modified ones did not. Under high temperature and pressure hydrothermal conditions, the amino groups are very easy to interact with carboxyl ones, which can form acylamino so that the primary amine group is hardly left [69]. Combined with the XPS spectra, the successful modification of amino groups into CD and spCD to form NH_2_–CD and NH_2_–spCD, respectively, could be confirmed.

### 3.4. Bacterial Imaging by Carbon Dots

#### 3.4.1. Imaging for Dead Bacteria

As suggested, a smaller size of 5 nm would be good at penetrating bacterial cells at the same cultural time [13,14,15,16,60] and other functional groups should also be determining factors. As a cationic surfactant, benzalkonium bromide (BZ) was generally used as a bactericide for surface sterilization through the electrostatic interaction [22,37]. As Appendix A show, all types of carbon dots could stain bacterial cells without a difference regardless of gram types, which may be due to the electronic attraction between the negatively charged carbon dots and the bacterial and positively charged benzalkonium bromide on the dead bacteria [14,41,62]. Considering the negative charge of the four types of carbon dots (Figure 3d), electrostatic adsorption would not be the main reason for their staining ability. The negative charge of the four carbon dots can be confirmed by characterizing the zeta potentials (Figure 3c). We also studied the the zeta potential change of the bacteria after being cultured with benzalkonium bromide and subsequently being cultured with the carbon dots. As shown in Figure 3c, the surface charges of the bacteria moved toward the positive trend, which then turned to a negative change upon further incubation with carbon dots. The results indicated that four carbon dots can combine with the residual groups of benzalkonium bromide on a bacterial surface on the BZ-treated bacteria, which can thus be used for the imaging of dead bacterial cells (Appendix A).

Since bacterial death was not always under drastic circumstances, for example, killing by bactericides, the 60 °C one-hour water bath treatment as a mild method to obtain dead bacteria was employed, which would not significantly damage the bacteria membrane like benzalkonium bromide [24]. According to the theory mentioned in the first paragraph of Section 3.4.1, a size under 5 nm is an essential condition for nanomaterials to enter the bacteria. However, the imaging behaviors were different among the four types of carbon dots. As Figure 5 shows, spCD could more dramatically dye the dead cells than CD, and NH_2_–spCD could achieve more efficient dying compared with NH_2_–CD (Appendix A). The possible reasons are summarized as follows: One is the QY difference between spermine-integrated ones (CD and NH_2_–CD for 48.79% and 40.42%, respectively) and non-spermine-integrated ones (spCD and NH_2_–spCD for 66.46% and 53.47%, respectively), accordingly, higher QY would be favorable for fluorescent imaging. The other is that the surface functional groups on the carbon dots could also influence the imaging efficiency. As shown in Figure 4, the high-content of the C–O peak was observed in the spermine-integrated ones (spCD and NH_2_–spCD, Figure 4n,p) compared with the non-spermine-integrated ones (CD and NH_2_–CD, Figure 4m,o), whilst the former presented the C–O peak at high-resolution XPS which the latter did not show. Considering all types of CQDs have carboxyl groups, there might be more carboxyl on the surface of the spermine-integrated ones to influence the affinity toward bacteria [20,54]. In addition, the four types of carbon dots do not differ much in size. As a result, small size would not be the only important factor to influence bacterial imaging. The staining results of NH_2_–CD and NH_2_–spCD are also carried out in Appendix A, which reveals a better staining performance of NH_2_–spCD than NH_2_–CD. This may also be due to the smaller size and higher QY of NH_2_–spCD than NH_2_–CD.

#### 3.4.2. Live Bacterial Imaging

Compared with the staining of dead bacterial cells, live ones are more difficult to dye because of the dense bacterial cell walls and the charge barrier deriving from chemical groups on their cell walls [14,15,16]. Many kinds of antibacterial medicines or antibiotics have amino or acylamino groups that could be chosen to use on the surface of bacteria or could enter the inside of bacteria such as β-Lactam or sulfonamide [30,31]. As a result, precursors containing amido were considered for synthesizing carbon dots including ethylenediamine and spermine. Though acylamino and secondary amino were represented after a one-step hydrothermal method, both CD and spCD failed to stain alive bacterial cells as Figure 6 depicts. Since previous research has indicated that the primary amine group has an affinity towards the bacterial surface [26,32,34], ethylenediamine was then applied to modify the carbon dots on their surface to obtain NH_2_–CD and NH_2_–spCD (with primary amine groups). As expected, both *E.coli* and *S.aureus* were stained successfully by NH_2_–spCD (Figure 6). However, the result of NH_2_–CD was not satisfied for both *E.coli* and *S.aureus* imaging (Appendix A), which should be due to the shallow fluorescent comparing with NH_2_–spCD. The characterization of carbon dots in Figure 3a has proved our concept: the peak location and strength of CD and spCD were nearly identical as well as NH_2_–CD and NH_2_–spCD, which may be because they have the same precursors for synthesis. However, the modified groups (NH_2_–CD and NH_2_–spCD) presented primary amine while the non-modified groups (CD and spCD) did not. XPS further confirmed the more primary amine groups on the surface of the spermine-integrated ones (NH_2_–CD and NH_2_–spCD). In Figure 6, primary amino functionalized NH_2_–spCD successfully stained the alive bacteria, which indicates that the primary amine groups on the surface of carbon dots is a significant factor for staining alive bacteria in addition to the factor of high QY. Table 1 shows a summary of the four types of CDs in aspects of imaging and other properties which may suggest that spermine could enhance the QY value and primary amine may promote the behavior of alive cell staining. Though the zeta potentials are not beneficial for bacterial imaging, the extremely small size of CDs may make CDs go through the cells towards the damaged cell walls [35].

#### 3.4.3. MTT Assay

An MTT assay was carried out for CD, spCD, NH_2_–CD and NH_2_–spCD. Concentrations of carbon dots were designed at 1, 10, 50, 100 μg/mL and 0 μg/mL as a control group. After the staining process, the relative absorbances were calculated. As described in Figure 3e, all types of carbon dots have no effect on L929 cell viability even under a concentration of 100 μg/mL, which suggests a great biocompatible property.

## 4. Conclusions

Four types of carbon dots were synthesized for imaging dead or alive bacterial cells. The imaging of dead cells was firstly investigated by four types of carbon dots. Dead bacterial cells treated with 3% benzalkonium bromide can be stained by all types of carbon dots, while the one treated with a 60 °C water bath for 1 h can be stained by only spermine–carbon dots (spCD and NH_2_–spCD). Spermine can enhance QY through nitrogen doping that causes a fluorescent enhancement for imaging, while the small size, with an average diameter of 1.38 nm, makes it possible for spCD to penetrate dead bacteria cell walls. In addition, more carboxyl groups may help spCD be close with cell walls because of the amidogen–carboxyl interaction between bacterial cell walls and the surface of spCD. For the next step, carbon dots were employed to stain alive bacteria. Only NH_2_–spCD gave out a reasonable result, which may be attributed to the modification of primary amine groups. Notably, primary amine groups were not found on the surface of CD and spCD. Therefore, this work provides a special angle onsuperficial groups of carbon dots for bacterial fluorescent staining. However, there are still several insufficient points. Because of the similar excitation–emission peak value and wide emission range, these carbon dots could not recognize dead or alive bacterial cells compared with some other studies [22,23,35]. The fluorescent intensity for imaging alive cells was not as strong as the imaging of dead ones, since NH_2_–spCD possesses a negative charge. Considering the high QY value, the primary amine groups may not the only reason for sticking to bacterial cells. Furthermore, the tissue imaging of bacterial cells would be realized if the carbon dots could realize two-photon fluorescence owning a near infrared region adsorption [70,71]. Other factors, such as the interactions of other chemical groups between carbon dots and bacteria and the effect of hydration particle size, should be discussed more specifically in order to better demonstrate the mechanism for bacterial imaging using carbon dots. Efforts in staining intensity and specificity need to be improved to create a live/dead kit to replace the expensive commercial dyes.

## Data Availability

The original data supporting reported results can be obtained through the e-mail (yhxu@qdu.edu.cn) for asking.

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
