# Peer review of "Primary Amine Functionalized Carbon Dots for Dead and Alive Bacterial Imaging"

_nanomaterials, 2023, doi:10.3390/nano13030437_

Round 1

Reviewer 1 Report

This is an excellent paper in which the authors developed a new type of CDs for bacterial imaging. The experiments are very designed and performed, and the results are convincing. I would like to recommend the publication of the work in this journal after minor revision. Some comments:

The title: “Primary amino” should be “Primary amine”

The authors are suggested to draw a scheme to show how the CDs were fabricated.

Some related references on CDs and bacterial imaging should be mentioned in the Introduction section: for example: Anal. Chem. 2022, 94(10), 4243−4251.

Ref. 22 and Ref. 48 are the same. Please revise.

Ref. 15 and Ref. 61 are the same. Please revise.

The advantages and disadvantages of the CDs compared with those in previous studies that can also realize bacterial imaging should be discussed.

Author Response

We appreciate your advice which is very constructive. We carefully revised our manuscript with yellow highlighted in the content. And we hope to meet the requirments. 

Q1. The title: “Primary amino” should be “Primary amine”

 A1. Thank you for your advice. “Primary amino” has been changed into “primary amine” in the manuscript, the changes were red-colored.

Q2. The authors are suggested to draw a scheme to show how the CDs were fabricated.

 A2. A scheme for description of fabrication was added below the ‘introduction’ text, the contents were added in red color.

Q3. Some related references on CDs and bacterial imaging should be mentioned in the Introduction section: for example: Anal. Chem. 2022, 94(10), 4243−4251.

A3. We appreciate the reviewer’s recommendation. The reference has been added into introduction section where the marker is yellow highlighted.

Q4. Ref. 22 and Ref. 48 are the same. Please revise.

A4. Thank you for the reviewer’s suggestion. The mistake has been corrected.

Q5. Ref. 15 and Ref. 61 are the same. Please revise.

A5. Thank you for the reviewer’s suggestion. The mistake has been corrected.

  1. The advantages and disadvantages of the CDs compared with those in previous studies that can also realize bacterial imaging should be discussed.

A6. Thanks for your advice. The sentence for discussion has been added in “conclusion“, and the discussion were yellow highlighted.

Reviewer 2 Report

In this manuscript, the authors explore the use of carbon dots in the task of  bacteria imaging. Many aspects of preparation, characterization and imaging are described in details. While I do not see any fundamental novelty I think this manuscript could be treated as a good reference for applications. In this regard I think some minor questions should be addressed before I recommend this manuscript for publication.

(1) Quantum efficiency is not the only important value that characterises fluorescent label. If one wants to estimate brightness of this labels one should know some other numbers like absorption crossection and maximum count rate. Some additional information about represented carbon dots should be given or discussed.

(2) When authors write about absorbance carbon dots suspension how do they take into account scattering on carbon dots? Please provide estimation about how much absorption is greater than scattering in your experiment 

(3) Many figures are terribly represented. The font size looks almost 10 times different (for example, the captions  of the axes in Figure 2 e) And the graph to the right of Figure 2(e) is not described at all

Author Response

We appreciate your advice which is very constructive. We carefully revised our manuscript with yellow highlighted in the content. And we hope to meet the requirments. 

Response to Reviewer 2

Q1. Quantum efficiency is not the only important value that characterises fluorescent label. If one wants to estimate brightness of this labels one should know some other numbers like absorption crossection and maximum count rate. Some additional information about represented carbon dots should be given or discussed.

A1. I appreciate your advice. In addition to quantum efficiency, the absorption crossection and maximum count rate are important to evaluate a fluorescence probe. However, we use bacterial cells for our application aims, whose volume and contrast are much smaller or lower than animal and plant cells. Under 10 x eyepiece plus 40 x objective lens, there are only small points could be seen, and the resolution ratio of our fluorescent microscopy could not supply for more elaborate imaging. Actually, the original pictures shows that the probe could stain almost every cell if the probes are successfully stained the bacterial cells. The differences are the fluorescence intensity and the clarity of cell boundaries among different probes. As a result, we would only judge the imaging effect by the contrast between the background (nearly black) and the intensity of fluorescent signals in a crude way. Besides, cells including bacterial cells are confined space comparing to the whole solution containing bacterial cells. The fluorescent probe left in the solution could also be excited by mercury lamp to form the background. If the probes were not concentrated on the confined space which are bacterial cells, the bacterial shape of fluorescence signals won’t be shown. As long as the contrast of background and fluorescence signals are distinguished strongly, the bacterial cells would be seen and displayed. In some cases, although the fluorescent could be recognized for carefully observed (Figure 6 E.coli by spCD), the fluorescence intensity was far weak than the NH2-spCD exhibited. We are sorry for not calculating the absorption crossection and maximum count rate.

Q2. When authors write about absorbance carbon dots suspension how do they take into account scattering on carbon dots? Please provide estimation about how much absorption is greater than scattering in your experiment 

A2. Thanks for your advice. There is a Rayleigh scattering at around 350 nm when the emission scan was carried out. But the intensity of Rayleigh scattering is pretty weak (the peak usually at 1×102~3 counts) in contrast with the carbon fluorescence emission (the peak usually beyond 106). Actually, Rayleigh scattering signal would be neglected.

Q3. Many figures are terribly represented. The font size looks almost 10 times different (for example, the captions  of the axes in Figure 2 e) And the graph to the right of Figure 2(e) is not described at all.

A3. Thanks for your kindly advice. The figures have been remodified to be presented with consistent front. In addition, the captions of the axes in Figure 2 e) and the And the graph to the right of Figure 2(e) have been modified and described correctly. 

Reviewer 3 Report

The paper «Primary amino functionalized carbon dots for dead and alive bacterial imaging by Y.Liu, D.Zhong, L.Yu, Y. Shi, Y.Xu» is devoted to investigation of new types of carbon dots (CD)  for bio-imaging application, in particular, for imaging of bacteria.

The authors have done a great job both in creating the modifications of carbon quantum dots and in their comprehensive research. The text contains very detailed descriptions of the fabrication procedure as well as measurement techniques and procedures using a wide range of equipment and methods. Such a multimodal and complex approach deserves high praise, and, in general, I believe that the material presented in the work deserves publication.

However, I would also like to make a few remarks.

First, it concerns a review of the application of carbon nanodots for imaging,  I would like to recommend to present short comparison with other particles used for bioimaging: such as silicon, metal, conjugatred polymers nanoparticles, etc.

At present, particles and dyes are widely developed for obtaining images in the deep layers of biological tissues by nonlinear optical methods. Therefore, that would be nice a brief discussion of the possibilities of these particles for realizing two and three-photon fluorescence.

The main remarks are related to the presentation of imaging results, both dead and live cultures of bacteria

- At present text, the analysis and interpretation of the imaging is rather crude and conjectural in terms of staining features of the developing types of CDs. I would like to recommend the authors to summarize the results in the form of a table (tables), where, in addition to the particle size, their quantum yields, also indicate the quantitative imaging results (both in terms of fluorescence intensity and the number of stained bacteria in culture). I think that this will help more definitely highlight trends in the modification of quantum dots by spermine and/or amino groups.

- What are the statistics of experiments in bacterial cultures? Is this the only implementation?

- Figure 2b presents one of the central results of the work connected with size and quantum yield of CD. It presents a large number of references to the work of other authors, which it would be better to highlight specifically (Maybe, to make Fig.2b as a separate pictute)

Thus, I repeat once again that I would like to support the publication of work after the major revision, related to the presentation and interpretation of imaging results.

Author Response

We appreciate your advice which is very constructive. We carefully revised our manuscript with yellow highlighted in the content. And we hope to meet the requirments. 

Response to Reviewer 3

Q1. It concerns a review of the application of carbon nanodots for imaging,  I would like to recommend to present short comparison with other particles used for bioimaging: such as silicon, metal, conjugatred polymers nanoparticles, etc.

A1. I appreciate your advice. The short discussion has been added into “introduction section” at the end of second paragraph with yellow highlighting.

Q2. At present, particles and dyes are widely developed for obtaining images in the deep layers of biological tissues by nonlinear optical methods. Therefore, that would be nice a brief discussion of the possibilities of these particles for realizing two and three-photon fluorescence.

A2. Thanks for your advice. Generally, to realize the two and three-photon fluorescence,  near infrared region (NIR, 1000~1700 nm) emissions especially NIR-â…¡(Ref 10.3389/fchem.2021.689017 or 10.1002/smll.202002054.) was usually needed for the fluorescence probe, quantum dots. Usually, carbon dots owning ultraviolet or visible light wavelength emissions (around 300~700 nm) could not realize the tissue penetrating imaging. Carbon dots with NIR emissions are rarely reported, but show promising potential for two and three-photon fluorescence. So the comment of the reviewer is valuable for the future development, which has been added in Conclusion section as yellow-highlighted.

Q3. At present text, the analysis and interpretation of the imaging is rather crude and conjectural in terms of staining features of the developing types of CDs. I would like to recommend the authors to summarize the results in the form of a table (tables), where, in addition to the particle size, their quantum yields, also indicate the quantitative imaging results (both in terms of fluorescence intensity and the number of stained bacteria in culture). I think that this will help more definitely highlight trends in the modification of quantum dots by spermine and/or amino groups.

A3. I appreciate your advice. Table 1 and corresponding explanations was added at the end of 3.4.3. with yellow highlighting

Q4. What are the statistics of experiments in bacterial cultures? Is this the only implementation?

A4. Thanks the reviewer’s comment. Every experiment was conducted by at least 3 times to assure the reproducibility and reliability of the results.

Q5. Figure 2b presents one of the central results of the work connected with size and quantum yield of CD. It presents a large number of reference to the work of other authors, which it would be better to highlight specifically (Maybe, to make Fig.2b as a separate pictute)

A5. We appreciate your advice. The purpose of figure 2b is to exhibit the small size with reasonable quantum yield which would be a promising fluorescence probes comparing with other researches. However, considering the type settings of all figures, we decide to amplify the font size in order to make the graph clearer.

Round 2

Reviewer 3 Report

I would like to express my satisfaction with the work done by the authors of the article and the changes made. I think that in this form the flock can be published.